# Placental Angiodysplasia: A New Sign for Prediction of Fetal Outcome?

**DOI:** 10.3390/jcm12113835

**Published:** 2023-06-03

**Authors:** Andrea Marzullo, Emmanuela Vitelli, Gerardo Cazzato, Margherita Fanelli, Giuseppe Ingravallo, Antonella Vimercati, Roberta Rossi, Leonardo Resta

**Affiliations:** 1Section of Molecular Pathology, Department of Precision and Regenerative Medicine and Ionian Area (DiMePRe-J), University of Bari “Aldo Moro”, 70124 Bari, Italy; andrea.marzullo@uniba.it (A.M.); e.vitelli@studenti.uniba.it (E.V.); margherita.fanelli@uniba.it (M.F.); roberta.rossi@policlinico.ba.it (R.R.);; 2Section of Gynaecology and Obstetrics, Department of Precision and Regenerative Medicine and Ionian Area (DiMePRe-J), University of Bari “Aldo Moro”, 70124 Bari, Italy; antonella.vimercati@uniba.it

**Keywords:** placenta, angiodysplasia, malformation, outcome, neonatal, obstetrics, gynaecology

## Abstract

The study of the placenta is of great importance, not only in the attempt to understand the etiopathogenesis of various maternal-fetal pathologies, but also in the attempt to understand whether it is possible to find the cause of pathological neonatal outcomes. On the other hand, abnormalities of blood vessel formation, such as angiodysplasias, have been poorly characterised in the literature, and there is a need for more studies investigating the potential impact on the fetus. In this paper, we retrospectively analysed 2063 placentas received at the Department of Pathology of the University of Bari ‘Aldo Moro’, among which we identified 70 placentas affected by angiodysplasia. On these placentas, we carried out histochemical staining with Masson’s Trichrome, orcein-alcian blue, and, subsequently, immunostaining with anti-CD31, CD34, and desmin and actin muscle smoothness antibodies. Finally, we performed a morphometric analysis on the allantochorionic and truncal vessels and correlated the results with neonatal outcomes. We studied the characteristics of the angiodysplasias in detail, dividing the patients into two classes (A and B) according to the morphology and histochemical characteristics of the affected vessels; statistical analysis reported a statistically significant association (*p* < 0.05) between the ratio of maximum thickness to maximum diameter (Tmax/Dmax) and neonatal outcome, with only 30% physiological outcome in the cohort of the placentas affected by angiodysplasia. These results shed light on a rather neglected aspect in the 2015 Amsterdam Classification, as well as in the literature, and provided strong evidence that placental angiodysplasia is predictive of an increased likelihood of the pathological fetal outcome, while other factors remain in the field. Studies with larger case series and guidelines with more attention to these aspects are mandated to further investigate the predictive potential of this pathology.

## 1. Introduction

The placenta is a fetal organ that is fundamental for the sustenance of the unborn child, characterised by the genetic heritage inherited from the mother and father, and with vascularisation mainly originating from the fetus [1]. The study of the placenta can provide important information on perinatal diseases, e.g., of infectious nature, or late conditions of metabolic and psychophysical nature [1,2]. It also provides important information about any maternal pathological conditions that may influence its structure [3]. The term angiodysplasia is derived from the Greek words ‘angio’ (vessel) and ‘dysplasia’ (wrong tissue formation). This definition (abnormal development of vessels) was first used in 1978 by Welch CE et al. [4], who decreed angiodysplasia of the colon to be the vascular lesion underlying copious rectal or intermittent bleeding of unknown etiology and pathogenesis. The most frequent localizations are bladder, intestine, uterus and brain [5]. The nosographic classification of vascular anomalies is still today a source of considerable difficulty and controversy due to the heterogeneity of the clinical-pathological entities and the terminological confusion generated in the past by the multiple medical and popular definitions rooted in common usage [6]. The need to bring order to the numerous terms, synonyms, and eponyms used in the past and to speak a common scientific language has led to the search for an international classification that offers the clinician a simple and pragmatic tool in the recognition and management of the various vascular anomalies. At present, the classification approved by the ISSVA (International Society for the Study of Vascular Anomalies) during the 11th International Workshop held in Rome in 1996, constitutes a valid basic reference in the management of vascular anomalies [7]. This classification distinguishes hemangiomas and benign neonatal tumours with spontaneous involution from vascular malformations proper, consisting of morphological and structural alterations on a dysembryogenetic basis of various districts of the circulatory system. Vascular malformations can affect the truncular vessels (i.e., the large vessels) or extra-truncular vessels (i.e., their branches) [7]. Placental vascular diseases can be divided into two large families: pathologies of the vessels connecting the chorionic plate with the fetus and vascular pathologies of the chorionic plate.

The first group coincides essentially with pathologies of the vessels of the funiculus and the amniochorial vessels. The second, on the other hand, coincides with lesions of the vessels of the first-, second- and third-order main villi and the chorial vessels of the mature and immature intermediate villi and also the capillaries of the terminal villi [8].

Angiodysplasia can affect both the walls of veins and arteries. In the veins, ectasia of a circumscribed or extended tract may be observed, resulting in an increase in the calibre of the vessel, but thrombus formation of the wall may also be present in the chorionic plateau. These lesions can be responsible for a critical state, especially when associated with a subchorionic or perivascular connective tissue inflammatory state [7,8].

In this manuscript, we conducted a histopathological, histochemical and immunophenotypical study of a cohort of 70 placentas with angiodysplasia and related the results to pre-existing maternal pathology and neonatal outcome in an attempt to assess the exact pathogenetic significance of the described lesions. Furthermore, we conducted accurate measurements of pathological and healthy vessels and compared them in order to understand whether and how much the vessel alteration affects the placental circulation and possible neonatal outcomes.

## 2. Materials and Methods

In our Department of Pathological Anatomy, from 2016 to 2022, 2063 placentas were analysed by careful macroscopic examination, followed by sampling and reading under the light microscope.

In 70 cases, angiodysplasia of not only small calibre but also medium- and large-calibre vessels was found during the histological study. All clinical data of these patients were retrieved from the paper medical records and, when not available, were searched on the electronic database. All pregnancies were full-term (>37 weeks–42 weeks), and no history of cigarette smoking or medication use during pregnancy was reported.

Furthermore, there were no patients suffering from genetic and/or acquired diseases of the cardiovascular system.

These morphological data were evaluated in a double-blind manner by two pathologists with expertise in gynaecological pathology (A.M. and L.R.) and, for each case under study, the slide considered most significant was selected and subsequently subjected to a series of histochemical and ancillary techniques aimed to study the wall of the large- and medium-calibre vessels, the endothelium and the mesenchyme.

In order to evaluate the distribution of vessels, immunohistochemical staining with anti-CD31/PECAM-1 (clone 1A10, Leica, autostainer Bond, 1:500 dilution, Wetzlar, Germany), CD34 (QBEnd10, Dako-Agilent, 1:400 dilution), Smooth Muscle Actin (SMA) (Clone 1A4, Dako-Agilent, 1:500 dilution, Santa Clara, CA, USA) and Desmin (Clone DE-R-11, Leica, 1:100 dilution, Wetzlar, Germany) antibodies were performed with antigenic unmasking heat-induced citrate buffer epitope retrieval, at pH 6.

Furthermore, histochemical staining with trichrome and orcein-alcian blue at pH 1 was performed.

Structural changes in the vessels were assessed by analysing:-Alterations of the intima;-Alterations of the media (relative to the elastic and muscular component of the wall);-Global alterations (intima and media).

The adventitia tonaca does not have its own dignity in the placental vessels.

The pathological features reported outside the vascular alterations were noted in the anatomopathology report of the individual cases. The patient’s medical records provided us with information on any maternal pathology during pregnancy. The neonatology discharge form was used to collect information on the newborn’s outcome.

From the archives of the Neonatology Unit, the well-being evaluation forms and any early complications that occurred were consulted. On the basis of the neonatology cards, the case history was divided into 7 groups:Preterm births;SGA < 10th percentile;Malformations;Jaundice;Sepsis;Perinatal death;Physiological outcome.

The fetal malformations included in group IV were:-Retinopathy of premature (ROP);-Perviousness of the Botallo arterial duct (PDA);-Bronchodysplasia;-Tricuspid insufficiency;-DIV (interventricular defect);-DIA (interatrial defect);-Right atriomegaly;-Hydronephrosis;-Transposition of the great arteries (TGA);-Aortic coarctation;-Annular pancreas;-Abnormal origin of the right coronary artery.

### 2.1. Morphometric Analysis

Careful measurements were taken on the truncal and allantochorial vessels, comparing the values of healthy and pathological ones. Each vessel was measured precisely:-Maximum diameter (D_max);-Maximum thickness (T_max);-Minimum thickness (T_min).

### 2.2. Morphometric Statistical Analysis

For each placenta, measurements of D_max, T_max, and T_min of healthy and pathological vessels were evaluated. The ratio of maximum thickness to maximum diameter (T_max/D_max) was assessed as a possible indicator of damage. This measurement was subsequently classified into four classes (≤0.15, 0.16–0.24, 0.25–0.33, ≥0.34) according to quartiles.

The above vessel measurements were compared between pathological and non-pathological vessels using a paired Student *t*-test, as they were assessed on the same placenta.

Subsequently, a comparison was made between the vessel measurements, both pathological and non-pathological, obtained from the placentas of women with physiological birth outcomes and those from women with pathological neonatal outcomes; this comparison was performed using an unpaired Student *t*-test. Neonatal outcome was related to classes (A and B), Amsterdam criteria, maternal pathology and classes of T_max/D_max ratios using the chi-square test. Where necessary, the presence of a trend in the distributions was assessed using the Cochran–Armitage test.

## 3. Results

The number of placentas examined from 2016 to 2022 was 2063 and there were 70 cases (3.4%) in which angiodysplasias were found.

The analysis of histological preparations has allowed us to constitute a new classification in which placental angiodysplasias have been divided into two classes, class A and class B.

Class A is characterised by fibro-leio-muscular hyperplasia of the tonaca intima. This hyperplasia took the form of a mostly asymmetric concentric arrangement (crescent arrangement) sometimes coexisting with rarefaction of the muscle fibres of the middle tonaca. In other cases, the intimal thickening appeared as a protruding formation in the lumen and sometimes with calcium salt deposition and the prevalence of fibrosis. This anomaly was often associated with thrombosis of other vessels. This group included 38 placentas (Figure 1A).

Class B is characterised by distraction and sloughing of the middle tonaca with muscle fibres spaced by an edematous space more abundant in the peripheral portion. The course of the fibres did not preserve the concentric orientation, but often the fibres presented a helical or irregular course. The remaining 32 placentas fall into this second group (Figure 2A).

In many cases, both types A and B were present in the same case and the reference of the latter to group A or B was made on the basis of the prevailing morphological characteristics.

Trichrome staining according to Masson was of fundamental importance in studying the connective tissue and muscle fibres of the placental vessels. It was the technique that was most influential in the determination of the two classes A and B, in which it revealed fibro-leio-muscular hyperplasia of the tonaca intima in class A and delamination and degeneration of the tonaca media in class B (Figure 1B and Figure 2B). Histochemical analysis demonstrated the presence of an unusual collagenous component in both the fibro-intimal pads and between the muscle fibres of the media.

Orcein staining, which was intended to highlight the presence of elastic fibres, was negative in all the preparations examined. This result allowed us to conclude that in the venous and arterial vessels of this organ, the elastic component is not present and is reduced to a few elastic fibres at the fibro-intimal border of the arteries. In angiodysplastic vessels, this absence is maintained.

With regard to ancillary techniques, staining with the anti-CD34 antibody showed intense positivity in the endothelia of the intermediate and terminal villi. A similar but weaker positivity was shown for the same structures with the anti-CD31 antibody. These two stains made a strong contribution in highlighting the vessels within the villi but played no role in the study of placental angiodysplasias.

Smooth muscle actin (SMA) was strongly positive in vessels with a muscular component, i.e., the allantochorial and truncal vessels, making it possible to study the helical and/or oblique course of muscle fibres (Figure 1C and Figure 2C).

The anti-desmin antibody marked smooth muscle cells with a lower intensity than smooth muscle actin, confirming what has already been demonstrated by the anti-smooth muscle actin antibody.

Finally, we performed immunohistochemical investigations for CD68 (PG-M1) and Ki-67 to assess the distribution of macrophage in the area of angiodysplastic changes in the two classes (A and B) and to evaluate the potential difference of proliferation index at this level. In class A, CD-68 showed a tendency to be present at the level of the tonaca intima/medium of the vessels (Figure 3A), as well as in Hofbauer cells present in the villi, while in class B, CD-68 was positive only in Hofbauer cells with no presence at the level of the vessel wall (Figure 3B).

Ki-67 was completely negative in both classes A and B.

Subsequently, the distribution of the 70 placentas in the seven neonatal outcome groups was evaluated in relation to classes A and B as reported in Figure 4.

Another assessment carried out in the course of our study was the frequency of placental lesions, found during reporting, according to the classification drawn up in the 2015 Amsterdam Consensus Conference in placentas with a concomitant diagnosis of angiodysplasia (Figure 5). Secondly, the distribution of these placentas in class A and B was analysed. In this summary, a histological report may have led to the inclusion of the same placenta in more than one category due to the concomitance of several etiopathogenetic pictures. The distribution of placental lesions, according to Amsterdam Criteria, was not significantly different between placentas of class A and B (χ^2^ = 1.7 *p* > 0.05).

In the course of the study, we also analysed maternal pathologies in history, including both pathologies occurring during pregnancy, such as gestational diabetes and pre-eclampsia, as well as some peculiar situations during gestation, such as abnormalities of the funiculus and placental insertion. No association was observed in relation to the classification of placentas A and B (χ^2^ = 5.17 *p* > 0.05).

Maternal pathologies were distributed in the following way: 7 cases of gestational diabetes, 13 cases of preeclampsia, and 5 cases of placental abruption.

The graph in Figure 6 shows, for each placenta examined, the difference in the T_max/D_max ratio between healthy and pathological vessels.

A prevalence of differences of less than 0 was observed, showing that the T_max/D_max ratio was greater in pathological vessels than in healthy vessels. The mean difference was statistically significantly different from 0 (t = 2.19, *p* = 0.03).

Classifying the neonatal outcome as pathological or physiological, we observed that among our sample of 70 placentas with angiodysplasia, only 30% had a physiological outcome.

The distribution of physiological and pathological outcomes in relation to classes A and B has been reported in Figure 7.

The data showed that 23.7% of class A angiodysplasia cases had a physiological neonatal outcome, while 37.5% of class A cases had a pathological outcome.

With regard to class B, 76.3% of the cases had a physiological neonatal outcome, with 62.5% of the cases presenting a pathological outcome. From all this, there is no significant difference in outcomes in relation to classes (χ^2^ = 1.58 *p* > 0.05).

The association between outcome and maternal pathology, on the other hand, was statistically significant (χ^2^ = 6 *p* = 0.014), as shown in Figure 8.

In the absence of maternal pathology, 87.7% of physiological outcomes were observed, against 55.1% of pathological outcomes. In the presence of maternal pathology, 14.3% physiological outcome was observed against 44.9% pathological outcome.

With regard to the morphometric analysis performed, as previously described in statistical terms, we related the value of the Tmax/Dmax ratio (divided into quartiles) to the neonatal outcomes. This is shown in the following graph (Figure 9).

A highly significant trend emerges from the data (Cochran–Armitage Trend Test = −4.72, *p* < 0.0001): as the absolute value of the Tmax/Dmax ratio increases, the physiological outcome decreases, from 52.4% in the first quartile, 33.3% in the second quartile, 14.3% in the third quartile to 0.0% in the last quartile, respectively. Consensually, the pathological outcome increases from 12.2% in the first quartile, 14.3% in the second quartile, and 32.7% in the third quartile to 40.8% in the last quartile, respectively.

As a confirmation, a significant difference in the T_max/D_max ratio between physiological and pathological outcomes was observed (t = −6.95 *p* < 0.0001) (Figure 10).

## 4. Discussion

The analysis of the Amsterdam classification drawn up in the 2015 Consensus Conference led us to highlight a lack of attention paid to certain vascular alterations of the amniochorial and umbilical vessels [9]. In fact, in the current literature, an important part concerning the pathology of fetal vessels, such as chorioangiomatosis, chorioangiomatosis and angiodysplasias, is little considered. These pathologies are of considerable importance both for the possibility of more frequent accidents coinciding with delivery and for the possibility of even serious repercussions on the dynamics of fetal circulation (fetal heart failure) [10].

We have detected alterations of the vessel wall, especially the arterial wall, not yet described in the literature, which consist of alterations attributable to vascular thrombosis and their evolution (categorised by us in Class A) and alterations concerning the distribution of the muscular fibres of the middle tonaca (categorised in Class B).

In the first case (Class A), we see how the semilunar or polypoid arrangement of the fibromuscular thickening of the intima is probably to be considered as the natural evolution of partial thrombosis of the arterial lumen. Both the correlation with thrombotic phenomena present in other vessels of the same placenta and the presence of regressive, necrotic changes and calcium deposition similar to what has been observed in adult atherosis, lead us to believe that these phenomena are consequent to intravascular coagulation. In the case of the allantochorial arteries, the resulting reduction of the lumen can easily be compensated for by openings of lateral anastomoses. In contrast, in the case of truncal artery involvement, the reduction of flow or obstruction is evidenced by the presence of fibrotic and avascular villi in its vicinity or by villous hypoplasia.

In the case of muscle dissociation (Class B), the vascular disturbance appears to affect both the reduction of the lumen and the possibility of reduced contraction force secondary to haemodynamic alterations resulting from the abnormal layering of muscle fibres.

The consequences on the dynamics of the placenta secondary to these alterations are not easy to deduce. While one can glimpse in some cases a dynamic that is not significantly influenced due to the focality and sectoriality of the phenomenon, one cannot overlook the fact that the alterations we have identified are correlated with other pathologies of a haemodynamic nature found in the placenta. In fact, in our study, the histopathological analysis of the placental preparations according to the Amsterdam classification [9] allowed us to identify 17 cases of early maternal malperfusion (8 belonging to class A and 9 to class B); 16 cases of late maternal malperfusion (8 belonging to class A and 8 to class B); 14 findings of chorioangiitis (9 belonging to class A and 5 to class B); and, finally, 14 placentas showing infectious phenomena of various kinds, from chorioamnionitis to funisitis (7 belonging to class A and 7 to class B).

Moreover, the health condition of the patients also often suggested the presence of possible altered haemodynamic conditions. In fact, the clinical history of the pregnant women allowed us to find 7 women with gestational diabetes (6 in class A and 1 in class B); 13 women with pre-eclampsia (7 in class A and 6 in class B); and 5 women with placental abruption (4 in class A and 1 in class B).

The correspondence of placental pathology with perinatal pathology is obviously preserved, as expected from normal knowledge of placental pathophysiology [11,12,13].

Looking at the pathology from the infant’s perspective, as we have done in this work, is certainly innovative and useful to understand the role of the placenta and its real impact on intrauterine life and the first fundamental days of extrauterine life [14].

In our case, we studied vascular alterations with undoubted repercussions on placental haemodynamics. Indeed, the physiological outcome in our pool of placentas with angiodysplasia was only 30%, demonstrating how the observation of aspects of angiodysplasia may constitute an important prognostic marker for the first moments of extrauterine life. The pathologies most frequently involved in children with placental angiodysplasia were prematurity, intrauterine growth retardation and malformations. Since the fetal vascularisation of the placenta mirrors the external vascularisation of the fetus, it may be assumed that, in addition to the reduction of maternal blood supply sometimes recorded, the haemodynamic restriction of fetal perfusion also affected fetal growth. In this sense, prematurity and growth retardation would find a solid explanation.

Malformations may also be the result of small- and medium-sized thromboses in the fetal circulation in the same way as in the placental vascular bed.

Our analysis of angiodysplasias has used traditional methods in the study of arterial and venous vessels.

Challier et al. in 2001 [15] identified CD31 and CD34 antibodies as two important markers of cell differentiation from the first trimester of pregnancy. Placental vessels develop and adapt with the function of supporting and nourishing the fetus. These two markers allowed us to carry out a precise study of the intermediate and terminal villi. In fact, the CD34 antibodies, in a more vivid way, and the CD31 antibodies, in a more attenuated way, allowed us to study the endothelium of the small-calibre vessels contained in the villi more marked than CD 31 [16]. However, these two antibodies made no significant contribution to the study of allantochorial and truncal vessels affected by angiodysplasia.

With regard to immunostaining with anti-desmin antibodies, in the slides we analysed we found positivity in the mesenchyme of the villi, allowing us to have an internal control of the samples examined, and in addition, we found focal positivity in the vessels affected by angiodysplasia.

With regard to the anti-smooth muscle actin antibody, in our work it proved to be an effective and somewhat irreplaceable marker of the muscular portion of the allantochorionic and truncular vessels, coming to the aid of cases in which it was difficult to interpret through histochemical techniques whether one or the other class of angiodysplastic vessels belonged to one or the other.

A role of fundamental importance in the study of placental angiodysplasias was played by Masson’s trichrome staining. Already, in the text published in 2017 by Malvasi, ‘Management and Therapy of Late Pregnancy Complications’, Resta et al. [3] stressed the importance of this histochemical method in the study of vasculopathies of the vessels connecting the chorionic plate to the fetus. This text also reiterated the need within elementary vascular lesions to distinguish between circumferential segmental lesions of the vessel wall, rather than concentric hypertrophy of the middle tonaca or a non-muscular thinning of the wall or degeneration of the middle tonaca. The same staining, in our case, allowed the detection first and classification later of vascular abnormalities of the allantochorial and truncal vessels.

With regard to the use of orcein, in our analysis its use excluded the absence of an elastic component in the angiodysplastic placental vessels, confirming the presence of rare elastic fibres ‘connecting’ the intima and the media tonaca.

Alcian blue reactions in placental tissue sections have been used since the 1980s [17]. The villi of term placentas with postpartum complications showed only a small proportion of glycosaminoglycans, as revealed by the alcian blue-stained preparations. In fact, fibronectin and type IV collagen represented the major component of the connective tissue of the villi and played an important role in the organisation of the structures of the placental villi. In the study of angiodysplasias, alcian blue highlighted the role of glycosaminoglycans and fibronectin in class B angiodysplasias, in which these two components supported the structural integrity of the delaminated vessels [18].

Morphometric investigation showed that the most severe alterations of fetal vessels occur in SGA, fetal malformations and preterm deliveries.

However, there is an important proportion of vascular alterations that are accompanied by physiological fetal outcomes. If we look at the average of the percentage values, we see a significant relationship between the deterioration of morphometric parameters and neonatal outcomes. These considerations lead us to believe that the presence of these morphological changes in the arterial vessels is a sign of an important pathological condition of the fetal vasculature, both at the placental level and probably at the fetal level.

It is therefore possible that preterm birth and reduced fetal weight may be considered consequences of a condition of not only maternal but also fetal malperfusion.

As far as fetal malformations are concerned, the issue is complex; indeed, the frequently observed cardiac malformation may be related to a condition of mesenchymal alteration that has caused both cardiac and vascular damage. On the other hand, however, it is possible that the presence of a cardiac malformation may affect the development and constitution of both intrafetal and placental vessels. The morphology of the changes we observed is very similar to that observed in adult subjects with arterial damage secondary to essential hypertension. The possibility that there may be arterial hypertension in the fetal period (also secondary to other factors such as cardiac malformations) cannot be excluded a priori.

## 5. Conclusions

We wanted to review a large number of cases of placental pathology received at our institute from 2016 to 2022 to verify the frequency of placental vascular anomalies of angiodysplastic nature and their incidence on neonatal outcome. From the observations made, we can conclude that:
-Some placental alterations frequently found during reporting (haemorrhage, chorioangiosis, vascular damage) are not sufficiently valued by the scheme proposed by the Amsterdam Consensus Conference of 2015, although in our opinion they seem to play an important role in some pregnancy pathologies;-The analysis of alterations in the vessel wall, especially the arterial wall, using histochemical and immunohistochemical methods, has made it possible to identify lesions not yet described in the literature and consisting of alterations attributable to vascular thrombosis and their evolution (Class A) and alterations concerning the distribution of muscle fibres in the middle tonaca (Class B);-The consequences on placental dynamics secondary to these alterations are not easy to deduce. While one can glimpse, in some cases, a dynamic that is not significantly affected due to the focality and sectoriality of the phenomenon, one cannot overlook the fact that the alterations we have identified are correlated with other pathologies of a haemodynamic nature found in the placenta;-The physiological outcome in our pool of placentas with angiodysplasia was 30%, demonstrating how the observation of aspects of angiodysplasia may constitute an important prognostic marker for the first moments of extrauterine life;-The pathologies most frequently involved in children with placental angiodysplasia were prematurity, intrauterine growth retardation, and malformations;-There are morphological alterations in the placental fetal circulation that may be considered analogous to those in adult essential hypertension;-These alterations are significantly correlated with fetal pathological outcomes.


The change of perspective in the evaluation of placental pathophysiology, not as the organ of the mother but as the life support of the fetus, offers a new perspective on the study of this important and ephemeral organ.

## Figures and Tables

**Figure 1 jcm-12-03835-f001:**
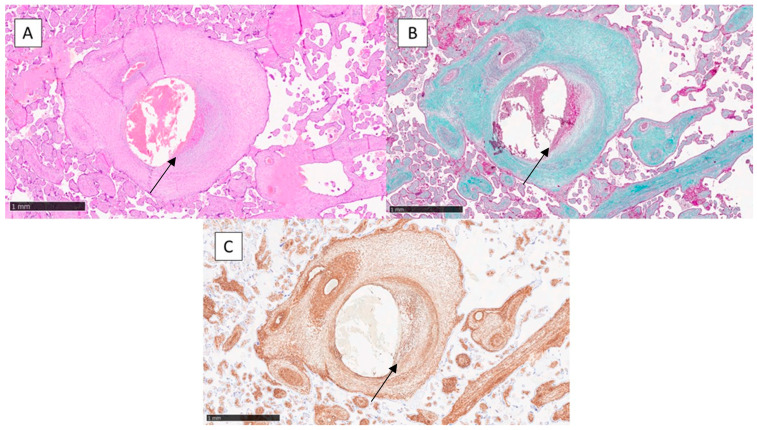
(**A**) Histological photomicrograph of serial sections showing arteries with fibro-intimal thickening and subsequent thrombosis in the organisation phase (black arrow), class A (Hematoxylin–Eosin, original magnification 10×). (**B**) Histochemical preparation showing the fibro-intimal thickening of the wall of the class A angiodysplasia and the reduction of lumen of the vessel (Masson’s trichrome, original magnification 10×). (**C**) Immunohistochemical preparation with anti-SMA antibody showing the presence of different orientations of the muscle fibres in class A angiodysplasia (Immunohistochemistry for SMA, original magnification 10×).

**Figure 2 jcm-12-03835-f002:**
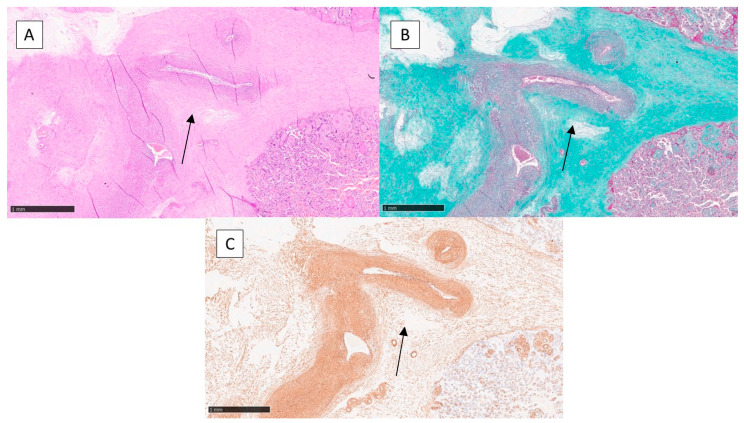
(**A**) Histological photomicrograph showing disarrangement of the muscle fibres of the middle tonaca of the arteries (black arrow), characteristic of class B (Hematoxylin–Eosin, original magnification 10×). (**B**) Histochemical preparation showing further disarrangement of the muscle fibres of the wall with aspects of initial arterial thrombosis (Masson’s trichrome, original magnification 10×). (**C**) Immunohistochemical preparation with anti-SMA antibody showing disarrangement of the muscle fibres of the wall and different types of orientation (black arrow) (Immunohistochemistry for SMA, original magnification 10×).

**Figure 3 jcm-12-03835-f003:**
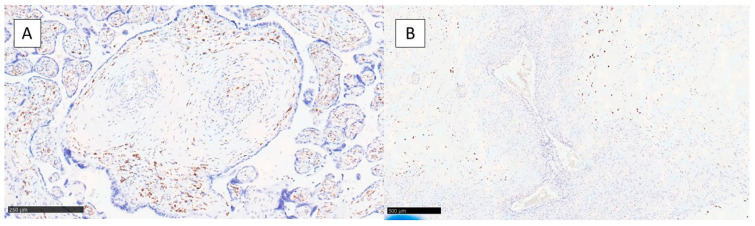
(**A**) CD-68 distribution in class A with positivity in the intimal/media tunica of the wall of the vessels. (**B**) Negativity for presence of macrophage in the wall of the vessels with presence of positivity in Hofbauer cells. (Immunohistochemistry for CD-68, original magnification 20×).

**Figure 4 jcm-12-03835-f004:**
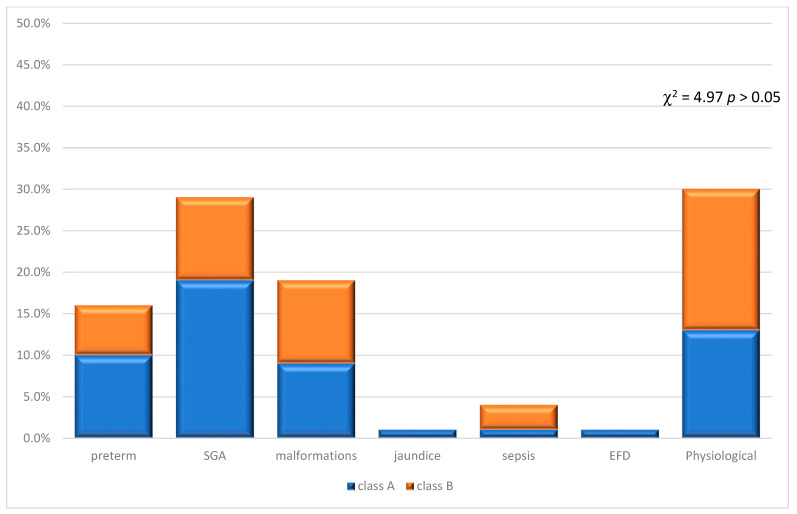
Neonatal outcomes in relation to classes A and B. The distribution of neonatal outcome resulted not associated with classes A and B (χ^2^ = 4.97 *p* > 0.05).

**Figure 5 jcm-12-03835-f005:**
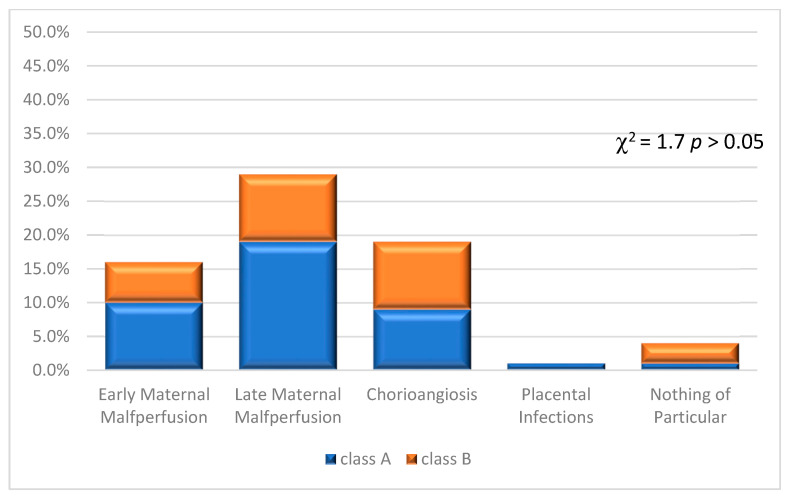
Distribution according to the 2015 Amsterdam Consensus Conference.

**Figure 6 jcm-12-03835-f006:**
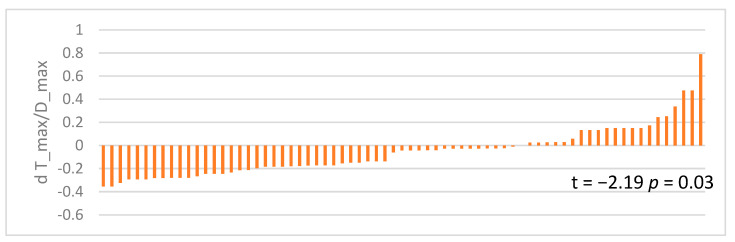
Difference in maximum thickness/maximum diameter ratios (δ mm) between healthy and pathological vessels for each placenta under study.

**Figure 7 jcm-12-03835-f007:**
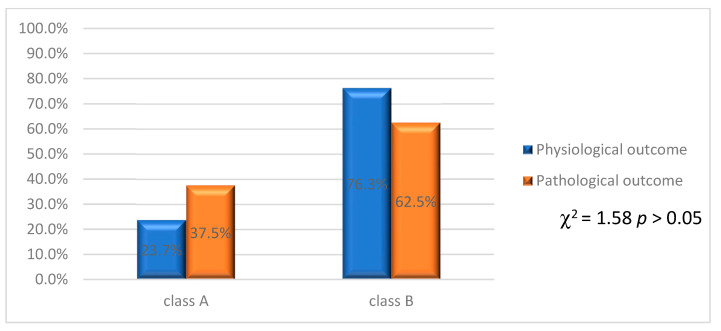
Physiological and pathological outcomes related to classes A and B.

**Figure 8 jcm-12-03835-f008:**
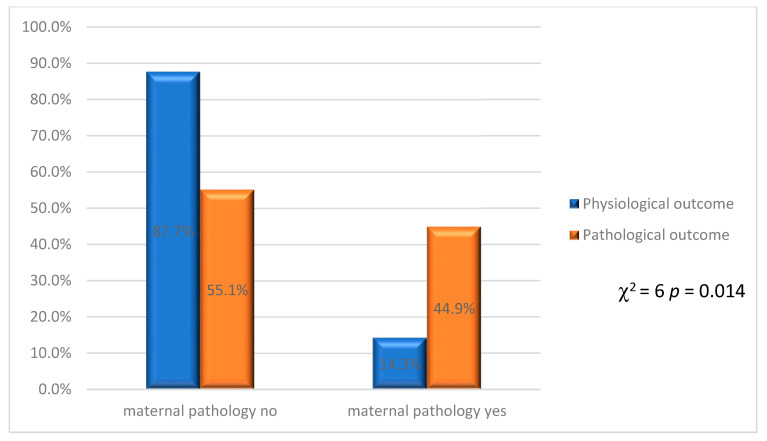
Physiological and pathological outcomes related to presence or absence of maternal pathologies.

**Figure 9 jcm-12-03835-f009:**
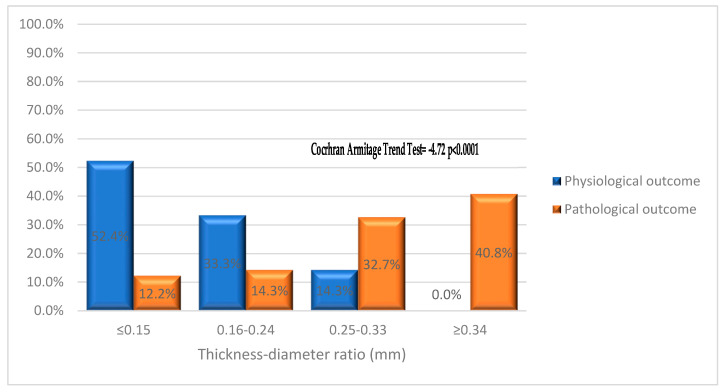
Neonatal outcomes in relation to the value of the T_max/D_max ratio.

**Figure 10 jcm-12-03835-f010:**
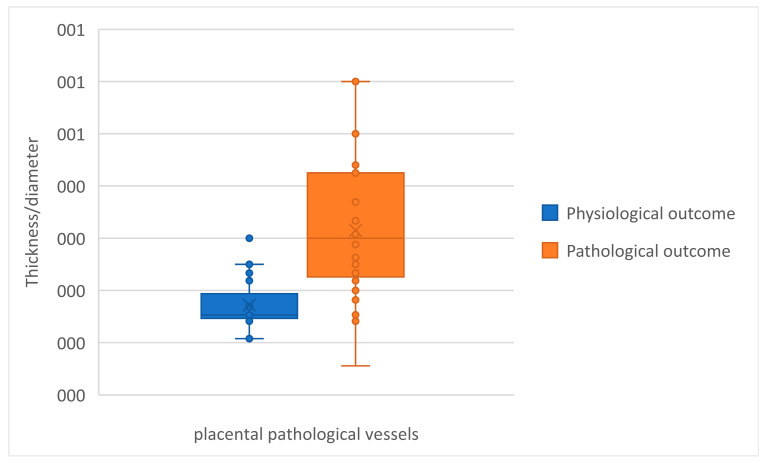
T_max/D_max ratio in pathological vessels in relation to physiological/pathological outcome.

## Data Availability

Not applicable.

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
