# Peer review of "Placental Angiodysplasia: A New Sign for Prediction of Fetal Outcome?"

_jcm, 2023, doi:10.3390/jcm12113835_

Round 1

Reviewer 1 Report

  Thank you for requesting  to provide a review of this article, which provides information about placental angiodysplazia, as a new sign for the prediction of fetal outcome .

   The main purpose of the analysis was to retrospectively analyse 2063 placentas received at the Department of Pathology of the University of Bari, and so it appeared that 70 placentas were affected by angiodysplazia. Morphogenetic analysis in the allantochorionic and truncal vessels in the placenta was made and the results were correlated with the neonatal outcomes. The main question adressed in the research was whether placental angiodysplazia can be a sign for maternal pathology and and neonatal outcome. Therefore, a histopathological, histochemical and immunophenotypical study of a cohort of 70 placentas was made, in attempt to assess the exact pathogenetic significance of the described lessions. Accurate measurements of pathological and healthy vessels were conducted and the results were compared in order to understand if and how much the vessel alteration affects the placental circulation and possible neonatal outcomes.

   The study is a cohort study. The topic is original and relevant in the field and brings usefull knowledge regarding the subject. A comprehensive search strategy was used and so, it appears that there is strong evidence that placental angiodysplazia is predictive of increased likelihood of pathological fetal outcome. It was demonstrated that the study of the placenta can provide important information on perinatal diseases and also about any maternal pathological conditions that may influence its structure. As described in the article, orcein staining, which was intended to highlight the presence of elastic fibres, was negative in all the preparations examined. As followes, in the venous and arterial vessels of this organ, the elastic component is not present and is reduced to a few elastic fibres at the fibro-intimal border of the arteries. The review methodology was comprehensive with screening and data extraction. When it comes to the methodology used, no specific improvements should be considered from my point of view.

   The conclusions are consistent with the evidence and the arguments presented, and they adress properly to the main question which conducted the analysis.

   The references are appropriate and well suited for this kind of study.

    Regarding the figures and pictures used in the article, they are very understandable and easy to be followed and they adress properly for this kind of study, so no other comments regarding this subject are necessary.

  Regarding the structure and accuracy of the phrases, the manuscript has well structured information, with supported evidence and well structured phrases.

   The manuscript is original and well defined. The results provide an advance in current knowledge. The results are being interpreted appropriately and are significant, as well as the conclusions.

  The study is correctly designed and the analysis is being performed at high standards, so the data are robust enough to draw the conclusion. Surely the paper will attract a wide readership.

   To conclude, the article is written in a proper way and brings useful information regarding the subject.

   I have some comments to add in the lines below, especially regarding the English language and the writting techniques, that needs to be improved a little.

Line 18: of the blood vessel, not „of blood vessel”

Line 19: fetus, not „foetus”

Line 30: of the placentas, not „of placentas”

Line 32: provided, not „provide”

Line 32: that the placental, not „that placental”

Line 32: predictive for, not „predictive of”

Line 39: The placenta, not „Placenta”

Line 41: fetus, not „foetus”

Line 49: ethiology, not „aetiology”

Line 51: heterogenity, not „heterogeneity”

Line 53: rooted in the, not „rooted in”

Line 59: of the, not „of”

Line 63: „Vascular malformations can affect the truncular vessels...”, not „Vascular malformations proper can be of the truncular vessels...” because the meaning of the phrase is not clear enough

Line 66: fetus, not „foetus”

Line 92: aimed to studying, not „aimed at studying”

Line 107: was divided, not „is divided”

Line 117: were, not „are”

Line 164: included, not „includes”

Line 173: edematous, not „oedematous”

Line 215: what has already been demonstrated, not „what had already been demonstrated”

Line 218: „.” after „Figure 5”

Line 250: „.” after „Figure 8”

Line 258: „.” after „Figure 9”

Line 281: „.” after „Figure 11”

Line 345: has used traditional methods, not „has used methods traditionally used”

Line 349: fetus, not „foetus”

Line 395: fetal, not „foetal”

Line 404: fetal, not „foetal”

Line 435: fetus, not „foetus”

Author Response

Dear Reviewer n'1,

thank you very much for your congratulations: we are very glad. Furthermore, we added and correct all in order to your suggestions.

Thanks again,

Reviewer 2 Report

The idea of the study is interesting, however in its present form there are major flaws and missing necessary parts of the study.

Concerns:

- study group is insufficiently presented/characterized. Term pregnancies? Any nicotinism, drugs during pregnancy (i.e. ASA, heparins) etc.? Exclusion criteria, e.g. mothers' genetic/vascular pathologies? - definition of SGA in line 109 -> below 2000g whereas now it is described as <10%; line 387 -> this is not true that this is growth retardation... Big mistake.

- malformations described in general as vascular pathologies, whereas not only?

- missing description of methodology in many parts! Only one sentence concerning IHC?! What Ab? dilutions, clones, pH, protocol, why exactly only those? What about Ki-67, CD-68 and others?

- how morphometry was performed? only 132-136? This is very important part of the paper and for sure analysis with the usage only HE topographically stained slides is deeply insufficient and non-convincing... There are methods of histochemical stainings which may be used, e.g. MOVAT...

- what is the origination of muscular hyperplasia in the tunica intima as there are originaly no muscle fibers in this regio? In this compartment exist epithelium and connective tissue with different proportions of fibers, whereas muscles are in tunica media. Line 139...?

- images are of weak quality and insuffcient magnification. Moreover there are missing arrows or asteriskc to point out iomportant parts of the sample.

- in such type of the study serial sections are in my opinion gold standard... I would expect presentation of used markers and methods in the form of panel of images. 

- Figure 5 presents only 7 groups while there is description of 8 groups. 

- why there is no presentation of all performed reactions/markers and methods? 

- wrong description of axis in the figure 11. 

There are plenty of amendments which should be introduced in the manuscript.

- minor language mistakes which can be easily fixed after deep reformulation of whole manuscript. i.a. 12.3% instead of 14.3%. line 267 or out of sense lin 247. 

Author Response

Reviewer n’2: Dear Reviewer n’2, first of all thank you very much for your kind suggestions. So, we have improved all parts of our manuscript following your valuables tips.

  • study group is insufficiently presented/characterized. Term pregnancies? Any nicotinism, drugs during pregnancy (i.e. ASA, heparins) etc.? Exclusion criteria, e.g. mothers' genetic/vascular pathologies?

Answer n’1: Thank you, we have added all available clinical informations about the pregnancies and potential drugs uses. Furthermore, we not stated as exclusion criteria the presence of genetic/vascular pathologies because we considered it to be implicit, but we have, quite rightly, specified it in materials and methods.

  • definition of SGA in line 109 -> below 2000g whereas now it is described as <10%; line 387 -> this is not true that this is growth retardation... Big mistake.

Answer n’2: Thank you, we have changed SGA definition and, also, we have corrected the big mistake: unfortunately, there was an error of writing.

  • malformations described in general as vascular pathologies, whereas not only?

Answer n’3: We have described all kinds of malformations. We hope that it will be fine.

  • missing description of methodology in many parts! Only one sentence concerning IHC?! What Ab? dilutions, clones, pH, protocol, why exactly only those? What about Ki-67, CD-68 and others.

Answer n’4: Thank you. We added all informations that you suggest us such as dilutions, clones, pH and antigen retrivieal protocol. We used these antibodies and not, for example Ki-67+, because it wasn’t important to asses the proliferative index or, for example, the macrophagic distribution (CD-68 PGM-1), but only the vessels distribution for the study of angiodysplasia. We added, also, a sentence speaking the reasons to use only these few antibodies. Thank you dear Reviewer n’2.

  • - how morphometry was performed? only 132-136? This is very important part of the paper and for sure analysis with the usage only HE topographically stained slides is deeply insufficient and non-convincing... There are methods of histochemical stainings which may be used, e.g. MOVAT...

Answer n’5: Dear Reviewer n’2, we understand that you may find our measurements on HE unconvincing, but we enclose an example of how the measurements were taken, taking, as is done in all morphometric measurements a sample of a truncal vessel and one of an allanto-chorionic vessel. This was carried out in a double-blind manner, so that the measurements could then be compared, using a semi-automated measuring system.

  • - what is the origination of muscular hyperplasia in the tunica intima as there are originaly no muscle fibers in this regio? In this compartment exist epithelium and connective tissue with different proportions of fibers, whereas muscles are in tunica media. Line 139...?

Answer n’6: Dear Reviewer, thank you for this question. Actually, the novelty of our paper lies precisely in the description of these morphological features: we found, through the investigations carried out, that there was a fibro-leiomuscular hyperplasia of the tunica intima exactly as occurs in the adult subject suffering from arterial hypertension; the histochemical stains made it possible to highlight the hyperplastic proliferation of smooth muscle cells at the level of the tunica intima due to a phenomenon of adaptation to stress (?)

  • - images are of weak quality and insuffcient magnification. Moreover there are missing arrows or asteriskc to point out important parts of the sample.

Answer n’7: Dear Reviewer, we are sorry to hear that the images are not of a high quality, as we acquired them with a professional Hamamatsu digital slide scanning system, and they are of a very high quality, having already published others in leading scientific journals. In any case, we have added some higher magnification images and indicative arrows. We hope it will be OK.

  • - in such type of the study serial sections are in my opinion gold standard... I would expect presentation of used markers and methods in the form of panel of images.

Answer n’8: Dear Reviewer, thank you very much. This is a paper that took us almost 6 years to sample all the placentas that came to the department. We did about 3 whole filing boxes with a total of more than 350 slides. For these reasons we decided that in the final paper we would include images as a "sample" of everything we processed over time, because it would have been impossible to upload panels of images due to the amount of data. We hope you can understand us. Thank you again.

  • Figure 5 presents only 7 groups while there is description of 8 groups.

Answer n’9: It’s correct, because the 8’ group is represented by physiological outcomes.

  • why there is no presentation of all performed reactions/markers and methods?

Answer n’10: We added all informations that you suggest us such as dilutions, clones, pH and antigen retrivieal protocol. We used these antibodies and not, for example Ki-67+, because it wasn’t important to asses the proliferative index or, for example, the macrophagic distribution (CD-68 PGM-1), but only the vessels distribution for the study of angiodysplasia. We added, also, a sentence speaking the reasons to use only these few antibodies. Thank you dear Reviewer n’2

  • wrong description of axis in the figure 11.

Answer n’11: Correct it. Thank you.

  • - minor language mistakes which can be easily fixed after deep reformulation of whole manuscript. i.a. 12.3% instead of 14.3%. line 267 or out of sense lin 247.

Answer n’12: Done, all correct. Thank you.

Reviewer 3 Report

a very well written papar on a really interesting topic.

The authors are clear in their formulation and structure of the manuscript, which runs like a red thread through the entire text.

Methods and results are well explained. The discussion is well conducted and openly and critically addresses the value of these discoveries.

3 points for improvement

1. Abbreviation for transposition of the great arteries is TGA. Please change!

2. Line 247-248 sentence is  incomplete

3. maternal pathology should be explained in a little more detail what exactly the authors mean by that. That doesn't really work

Author Response

Dear Reviewer n'3,

first of all thank you very much for your useful tips. 

We have changed following your precious suggestions:

1) We have correct the abbreviation TGA;

2) We have complete the sentence in lines 247-248;

3)We have added the clinical informations related to the maternal pathologies.

Thanks again for you time

Round 2

Reviewer 2 Report

There is improvement of submitted manuscript, however there are still important flaws.

                  missing description of methodology in many parts! Only one sentence concerning IHC?! What Ab? dilutions, clones, pH, protocol, why exactly only those? What about Ki-67, CD-68 and others.

Answer n’4: Thank you. We added all informations that you suggest us such as dilutions, clones, pH and antigen retrivieal protocol. We used these antibodies and not, for example Ki-67+, because it wasn’t important to asses the proliferative index or, for example, the macrophagic distribution (CD-68 PGM-1), but only the vessels distribution for the study of angiodysplasia. We added, also, a sentence speaking the reasons to use only these few antibodies. Thank you dear Reviewer n’2.

Rather in presented paper proliferation marker as well macrophages distribution IHC expression should be checked...

                  - how morphometry was performed? only 132-136? This is very important part of the paper and for sure analysis with the usage only HE topographically stained slides is deeply insufficient and non-convincing... There are methods of histochemical stainings which may be used, e.g. MOVAT...

Answer n’5: Dear Reviewer n’2, we understand that you may find our measurements on HE unconvincing, but we enclose an example of how the measurements were taken, taking, as is done in all morphometric measurements a sample of a truncal vessel and one of an allanto-chorionic vessel. This was carried out in a double-blind manner, so that the measurements could then be compared, using a semi-automated measuring system.

How exactly the Authors knew where are the boundaries on the HE image? This is non-convincing...

                 - what is the origination of muscular hyperplasia in the tunica intima as there are originaly no muscle fibers in this regio? In this compartment exist epithelium and connective tissue with different proportions of fibers, whereas muscles are in tunica media. Line 139...?

Answer n’6: Dear Reviewer, thank you for this question. Actually, the novelty of our paper lies precisely in the description of these morphological features: we found, through the investigations carried out, that there was a fibro-leiomuscular hyperplasia of the tunica intima exactly as occurs in the adult subject suffering from arterial hypertension; the histochemical stains made it possible to highlight the hyperplastic proliferation of smooth muscle cells at the level of the tunica intima due to a phenomenon of adaptation to stress (?)

Rather in presented paper proliferation marker as well macrophages distribution IHC expression should be checked... As mentioned above...

                 - images are of weak quality and insuffcient magnification. Moreover there are missing arrows or asteriskc to point out important parts of the sample.

Answer n’7: Dear Reviewer, we are sorry to hear that the images are not of a high quality, as we acquired them with a professional Hamamatsu digital slide scanning system, and they are of a very high quality, having already published others in leading scientific journals. In any case, we have added some higher magnification images and indicative arrows. We hope it will be OK.

Quality of IHC reactions is low, not quality of image precisely. Good previous publications are not guarantee of quality in present paper.

                 - in such type of the study serial sections are in my opinion gold standard... I would expect presentation of used markers and methods in the form of panel of images.

Answer n’8: Dear Reviewer, thank you very much. This is a paper that took us almost 6 years to sample all the placentas that came to the department. We did about 3 whole filing boxes with a total of more than 350 slides. For these reasons we decided that in the final paper we would include images as a "sample" of everything we processed over time, because it would have been impossible to upload panels of images due to the amount of data. We hope you can understand us. Thank you again.

What is an explanation? If there are that many samples it shouldn't be a problem to provide serial sections. At least panels of 4 IHC reactions may be used...

                 Figure 5 presents only 7 groups while there is description of 8 groups.

Answer n’9: It’s correct, because the 8’ group is represented by physiological outcomes.

That's not true. The Authors even stated that there are 8 groups.

                 wrong description of axis in the figure 11.

Answer n’11: Correct it. Thank you.

That's not true. Unchanged.

                 - minor language mistakes which can be easily fixed after deep reformulation of whole manuscript. i.a. 12.3% instead of 14.3%. line 267 or out of sense lin 247.

Answer n’12: Done, all correct. Thank you.

That's not true. Unchanged.

Author Response

Reviewer n’2: Rather in presented paper proliferation marker as well macrophages distribution IHC expression should be checked...

Answer n’1: Thank you. We performed (which is why it took us more than two weeks) immunohistochemical reactions for CD-68 (PG-M1) and Ki-67. We reported the data obtained in the major text of the paper.

Reviewer n’2: How exactly the Authors knew where are the boundaries on the HE image? This is non-convincing...

Answer n’2: We re-performed the measurements entirely on the Trichrome of Masson so that we could have a better account of the measurements with reliable separation of the various vessel structures. The results were completely superimposable.

Reviewer n’2: Rather in presented paper proliferation marker as well macrophages distribution IHC expression should be checked... As mentioned above...

Answer n’3: Thank you. We performed (which is why it took us more than two weeks) immunohistochemical reactions for CD-68 (PG-M1) and Ki-67. We reported the data obtained in the major text of the paper.

Reviewer n’2: Quality of IHC reactions is low, not quality of image precisely. Good previous publications are not guarantee of quality in present paper.

Answer n’4: We added serial immunohistochemistry images starting from the vessel structures studied in Hematoxylin-Eosin.

Reviewer n’2: What is an explanation? If there are that many samples it shouldn't be a problem to provide serial sections. At least panels of 4 IHC reactions may be used...

Answer n’5: We added serial immunohistochemistry images starting from the vessel structures studied in Hematoxylin-Eosin.

Reviewer n’2: Figure 5 presents only 7 groups while there is description of 8 groups.

Answer n’6: Thanks to your annotation we realized that we had reported a wrong classification with 8 groups, actually we corrected to 7 since, in the examined sample, there was no case of placenta with perinatal asphyxia outcome. Therefore, we have made appropriate corrections in the materials and methods and in the description of Figure 5.

Reviewer n’2:  wrong description of axis in the figure 11.

Answer n’7: We entered on the y-axis the thickness/diameter which is the evaluated ratio, on the x-axis the placental pathological vessels, in the legend we indicated the neonatal outcome corresponding to the placentas